# Deleterious neurological impact of diagnostic delay in immune-mediated thrombotic thrombocytopenic purpura

**Arthur Renaud** [1]*, **Aurélie Caristan**[2], **Amélie Seguin**[3], **Christian Agard**[1], **Gauthier Blonz**[4], **Emmanuel Canet**[3], **Marion Eveillard**[5], **Pascal Godmer**[6], **Julie Graveleau**[7], **Marie Lecouffe-Desprets**[8], **Hervé Maisonneuve**[2], **François Perrin**[7], **Mohamed Hamidou**[1], **Antoine Néel**[1]

1 Internal Medicine Department, Nantes University Hospital, Nantes, France, 2 Onco-Hematology and Internal Medicine Department, Departmental Hospital Center, La Roche-Sur-Yon, France, 3 Medical Intensive Care Unit, Nantes University Hospital, Nantes, France, 4 Intensive Care Unit, Departmental Hospital Center, La Roche-Sur-Yon, France, 5 Hematology–Cytology Department, Nantes University Hospital, Nantes, France, 6 Hematology and Internal Medicine Department, Hospital Center Bretagne Atlantique, Vannes, France, 7 Internal Medicine Department, General Hospital Center, Saint-Nazaire, France, 8 Internal Medicine Department, General Hospital Center, Cholet, France

* arthur.renaud49@gmail.com

## Abstract

### Background

Immune-mediated thrombotic thrombocytopenic purpura (iTTP) is a rare life-threatening thrombotic microangiopathy requiring urgent therapeutic plasma exchange (TPE). However, the exact impact of a slight delay in TPE initiation on the subsequent patients' outcome is still controversial.

### Aim

We aimed to study the frequency, short-term neurological consequences, and determinants of diagnostic delay in iTTP.

### Methods

We conducted a retrospective monocentric study including patients with a first acute episode of iTTP (2005–2020) classified into 2 groups: delayed (>24h from first hospital visit, group 1) and immediate diagnosis (≤24h, group 2).

### Results

Among 42 evaluated patients, 38 were included. Eighteen cases (47%) had a delayed diagnosis (median: 5 days). The main misdiagnosis was immune thrombocytopenia (67%). The mortality rate was 5% (1 death in each group). Neurological events (stroke/TIA, seizure, altered mental status) occurred in 67% vs 30% patients in group 1 and 2, respectively (p = 0.04). Two patients in group 1 exhibited neurological sequelae. The hospital length of stay was longer in group 1 (p = 0.02). At the first hospital evaluation, potential alternative causes of thrombocytopenia were more prevalent in group 1 (33% vs 5%, p = 0.04). Anemia was

**Data Availability Statement:** All relevant data are within the paper and its Supporting Information files.

**Funding:** The authors received no specific funding for this work.

**Competing interests:** The authors have declared that no competing interests exist.

less frequent in group 1 (67% vs 95%, p = 0.04). All patients had undetectable haptoglobin levels. By contrast, 26% of schistocytes counts were <1%, mostly in group 1 (62% vs 11%, p = 0.01).

## Conclusion

Diagnostic delay is highly prevalent in iTTP, with a significant impact on short-term neurological outcome. In patients with profound thrombocytopenia, the thorough search for signs of incipient organ dysfunction, systematic hemolysis workup, and proper interpretation of schistocytes count are the key elements of early diagnosis of TTP.

## Introduction

Immune-mediated thrombotic thrombocytopenic purpura (iTTP) is a thrombotic microangiopathy (TMA), a heterogeneous group of rare acute diseases characterized by peripheral thrombocytopenia, mechanical hemolytic anemia, and ischemic organ manifestations [1]. iTTP results from severe acquired ADAMTS13 (a Disintegrin And Metalloproteinase with ThromboSpondin-1 motifs, 13[th] member) deficiency, which leads to the accumulation of large Von Willebrand factor multimers, microthrombi formation, ischemic organ dysfunction, and hemolysis. Most iTTP cases exhibit anti-ADAMTS13 antibodies [2].

iTTP classical picture is a pentad of clinical-biological signs: severe thrombocytopenia, mechanical hemolytic anemia, fever, neurological involvement, and mild renal injury. However, only 5% of the patients exhibit all 5 signs initially [3]. Patients can present in the first place with nonspecific symptoms such as fatigue, headaches, nausea, vomiting or abdominal pain. Symptomatic cerebral and cardiac ischemia occurs in 40–60% and 10–15% of patients, respectively [4, 5], which almost invariably led to death in the absence of treatment, historically [6]. A suspicion of iTTP should prompt urgent plasma therapy using therapeutic plasma exchange (TPE) with fresh frozen plasma, pending diagnostic confirmation by ADAMTS13 activity measurement. Short-term mortality has dramatically improved and is now around 10% [7]. It may further improve with the advent of targeted therapies (i.e. caplacizumab) [8]. Modern data are scarce regarding the frequency, causes and consequences of delayed diagnosis in iTTP. Delayed TPE initiation has been associated with slower response to therapy and increased mortality in TMA as a whole [9, 10], but little data are available regarding iTTP itself. Two recent multicenter studies suggested that a slightly delayed diagnosis had no significant impact on mortality, but data regarding neurological outcomes are limited [11, 12].

The present study aimed to assess the frequency, neurological consequences and determinants of diagnostic delay in iTTP.

## Methods

### Patients

The cohort consists of patients admitted to our center (Internal Medicine Department, Nantes University Hospital) for a first acute episode of iTTP between 2005 and 2020. Their medical records were analyzed retrospectively. The confirmation of iTTP diagnosis rested on the association of signs of TMA (mechanical hemolytic anemia, acute thrombocytopenia, organ suffering) with no other causes identified, associated with ADAMTS13 deficiency <10%, and the presence of anti-ADAMTS13 auto-antibody or no persistent ADAMTS13 deficiency after treatment.

This study is in accordance with the Declaration of Helsinki, and the French Data Protection Authority and Legislation (MR003 reference methodology). No change in the current clinical practice and no randomization were performed. As it was a retrospective study, according to the French legislation (articles L.1121-1 paragraph 1 and R1121-2, Public health code), the head of the local ethic committee "Groupe Nantais d'Ethique dans le Domaine de la Santé" (GNEDS) confirmed that a formal review of the protocol by the ethic committee was not required.

## Data collection

Epidemiological data included first referral location, age at inclusion, sex, Charlson's comorbidity score, past medical history, and precipitating factors if identified. Clinical signs and organ involvements at first hospital visit have been registered, including neurological, chest, gastro-intestinal involvements, and hemorrhagic and general signs. Daily clinical notes provided any neurological events during hospital stay. Biological data—platelets count, hemoglobin level, reticulocytes count, lactate dehydrogenase (LDH) level, total bilirubin level, haptoglobin level, schistocytes count, and creatinine level with estimated glomerular filtration rate (eGFR)—were collected at first hospital evaluation and at the time of diagnosis. We also calculated the French score, a prediction score for TTP diagnosis in patient with TMA after ruling out intravascular disseminated coagulation, cancer, and solid organ/hematopoietic stem cells transplant situations. A French score of 2 (platelet count<30G/L and serum creatinine<200 μmol/L) is highly suggestive of TTP [13]. When available, results of bone marrow examination, anti-nuclear antibodies (ANA), anti-phospholipid antibodies (APL), and direct antiglobulin test (DAT) were also collected. The ADAMTS13 activity was determined by a method using commercial recombinant FRETS-VWF73 peptide, and anti-ADAMTS13 autoantibodies were screened using a commercial ELISA kit. All ADAMTS13 assays were performed in the Hematology Laboratory of Lariboisière Hospital, AP-HP, Paris. All the treatments administered, in an etiological purpose, were collected for each patient. We finally registered outcomes including first remission, ICU and hospital length of stay, neurological sequelae and death.

Organ involvements and clinical signs were defined as follows. Neurological involvement referred to isolated headache and neurological events. Neurological events encompass stroke, transient ischemic attack (TIA), seizure, and altered mental status. All patients that had a neurological event had at least a brain CT-scan. Chest involvement referred to the presence of chest pain, change in the electrocardiography measurement, elevation of troponin, heart failure, and/or heart rhythm disturbance. Gastrointestinal involvement referred to the presence of abdominal pain, nausea, and/or vomiting. Hemorrhagic signs were separated as cutaneous, mucosal or visceral. Constitutional signs included fatigue and fever. Remission was defined as a platelet count above 150,000/mm$^3$ for 48 hours with hemolysis resolution and organ-damage improvement or stabilization. The diagnosis was considered delayed when TPE initiation was ordered more than 24 hours after the first hospital visit.

Data were manually extracted from medical charts (AR and AC), containing patient identifying information, and anonymized before storage and data analysis.

## Statistical analysis

We described continuous variables as median with interquartile range, and categorical variables as percentages (%). Pearson's Chi$^2$ test with systematic Yale's correction assessed the differences between groups for categorical data. When the expected theoretical number in the contingency table was ≤5, we used Fisher's exact test. We employed the Mann-Whitney U test

to compare continuous data. To analyze neurological events and remissions over time, we drew survival curves using the Kaplan-Meier, and we compared the 2 groups using the log-rank test. The significance threshold used was $p<0.05$. We conducted the statistical analysis with GraphPad Prism software 6.0 (GraphPad Software, Inc., San Diego, California, USA).

## Results

### Diagnostic pathways and initial management

Between 2005 and 2020, 42 patients with newly diagnosed iTTP were managed at our center. We excluded 4 patients with incomplete medical charts and included the remaining 38 patients. Their main characteristics are shown in Table 1. The first hospital visit occurred at our institution in 39% of cases and another hospital in 61%. The median time between symptoms onset and the first hospital visit was 6.5 days (2–18). The first hospital visit occurred in an emergency department in 26 cases (68%), internal medicine in 5 (13%), hematology in 4 (11%), gynecology in 2 (5%) and cardiology in 1 (3%). Nineteen patients (50%) were referred to the hospital following the discovery of an abnormal complete blood count, 18 patients (47%) presented to the hospital for various symptoms which led to discover their hematologic abnormalities on site, and the remaining patient (3%) was diagnosed in the post-partum ward.

At first hospital visit, all patients were thrombocytopenic. Hemorrhagic features, present in 25 patients (66%), were mild or moderate (mostly purpura). The median hemoglobin level was 9.7 g/dL (8.9–11.3) and 7 patients (18%) had a normal hemoglobin level. Non hemorrhagic symptoms were present in 29 cases (76%): fatigue in 20 (53%), abdominal pain in 14 (37%), headache in 7 (18%), fever in 7 (18%), and/or nausea in 5 (13%). Overt brain and/or cardiac dysfunction were clinically apparent at first hospital visit in 6 cases (16%) each. Among 11 patients (29%) with an elevated troponin level, only 4 (11%) had clinical cardiac or ECG manifestations, or both.

Haptoglobin dosage, reticulocyte count, schistocytes count, lactate dehydrogenase, and troponin level were ordered at first hospital visit in 71%, 68%, 71%, 63% and 45% of cases, respectively. DAT was performed in 16 patients (42%) with only one (3%) positive, ANA in 33 (87%) with 15 (39%) above 1/80, and APL in 21 (55%) with 3 (8%) positive. ADAMTS13 activity was consistently below 10% and anti-ADAMTS13 antibodies were available for 29 patients (73%), positive in all cases.

Among 38 patients, 18 (47%) had a delayed diagnosis. Erroneous diagnoses were immune thrombocytopenia in 12/18 cases (67%), including Evan's syndrome in 6/18 (33%), heparin-induced thrombocytopenia in 2/18 (11%), and myelodysplasia, endocarditis, hemolytic uremic syndrome (HUS) and HELLP syndrome in 1/18 case each (6%). The frequency of diagnosis delay was consistent over time (2005–2009: 67%; 2010–2015: 42%; 2016–2020: 42%, $p = 0.41$).

Among 18 patients with delayed diagnosis, the median time between the first hospital visit and the diagnosis was 5 days (IQR 3–8, range: 2–51). Inappropriate measures to increase platelet count (platelet transfusion and/or thrombopoietin agonists) were used in 50% of these patients (versus 20%, $p = 0.09$). Eight patients (44%) received corticosteroids and 6 patients (33%) received intra-venous immunoglobulins before TPE initiation. Time course of hematological parameters is reported in Fig 1. Corticosteroids appeared to stabilize or improve platelet counts, but not hemoglobin levels.

### Deleterious neurological impact of diagnostic delay

To assess the clinical impact of diagnostic delay, we compared the clinical outcomes of the 2 groups. We found that as high as 67% of patients developed at least one neurological event in the delayed diagnosis group, compared to 30% in the other group ($p = 0.04$) (Fig 2A and 2B).

**Table 1. Characteristics of the study population at first hospital visit.**

| | Total (n = 38) | | Immediate diagnosis (n = 20) | | Delayed diagnosis (n = 18) | | P-Value |
|---|---|---|---|---|---|---|---|
| General | | | | | | | |
| Age–years (range) | 38 | (18–75) | 36 | (27–42) | 48 | (27–55) | 0.11 |
| Female sex–n (%) | 26 | (68) | 13 | (65) | 13 | (72) | 0.90 |
| Charlson score–points (IQR) | 0 | (0–1) | 0 | (0–0.25) | 1 | (0.25–1) | **0.005** |
| Underlying condition–n (%)[a] | 7 | (18) | 1 | (5) | 6 | (33) | **0.04** |
| Precipitating factor–n (%)[b] | 3 | (8) | 0 | (0) | 3 | (17) | 0.1 |
| Clinical manifestations | | | | | | | |
| Neurological involvement–n (%) | 10 | (26) | 5 | (25) | 5 | (28) | 1.0 |
| Headache–n (%) | 7 | (18) | 4 | (20) | 3 | (17) | 1.0 |
| Neurological event–n (%)[c] | 6 | (16) | 3 | (15) | 3 | (17) | 1.0 |
| Chest involvement–n (%) | 13 | (34) | 9 | (45) | 4 | (22) | 0.18 |
| Gastrointestinal signs–n (%) | 18 | (47) | 12 | (60) | 6 | (33) | 0.19 |
| Hemorrhagic signs–n (%) | 25 | (66) | 14 | (70) | 11 | (61) | 0.81 |
| Hemorrhagic score–points (IQR)[d] | 2.5 | (0–4) | 3 | (0–5) | 2 | (0–4) | 0.79 |
| Constitutional signs–n (%) | 23 | (61) | 12 | (60) | 11 | (61) | 1.0 |
| Fatigue–n (%) | 20 | (53) | 11 | (55) | 9 | (50) | 1.0 |
| Fever–n (%) | 7 | (18) | 2 | (10) | 5 | (28) | 0.22 |
| Biological features | | | | | | | |
| Platelet count–G/L (IQR) | 12 | (7–23) | 10 | (7–14) | 19 | (9–28) | **0.03** |
| Hemoglobin–g/dL (IQR) | 9.7 | (8.9–11.3) | 9.5 | (8.5–10.5) | 10.6 | (9–13) | 0.15 |
| Anemia–n (%)[e] | 31 | (82) | 19 | (95) | 12 | (67) | **0.04** |
| Reticulocytes–G/L (IQR)[f] | 143 | (89–222) | 143 | (87–217) | 142 | (108–214) | 0.82 |
| Schistocytes–% RBC (IQR)[f] | 1,6 | (0,8–4,0) | 1,8 | (1,3–4,2) | 0,6 | (0,5–1,5) | **0.03** |
| Schistocytes <1% RBC–n (%) | 7/27 | (26) | 2/19 | (11) | 5/8 | (63) | **0.01** |
| LDH–UI/L (IQR)[f] | 1207 | (719–1709) | 1231 | (852–1640) | 1182 | (480–1857) | 0.61 |
| Bilirubin–μmol/L (IQR)[f] | 34 | (24–52) | 36 | (32–55) | 29 | (19–43) | 0.15 |
| Haptoglobin <0.1 g/L–n (%) | 27/27 | (100) | 20/20 | (100) | 7/7 | (100) | 1.0 |
| eGFR–mL/min/1.73m$^2$ (IQR) | 77 | (56–101) | 63 | (51–99) | 85 | (72–108) | 0.11 |
| French score[g] | | | | | | | |
| 0 –n (%) | 0 | (0) | 0 | (0) | 0 | (0) | 1.0 |
| 1 –n (%) | 5 | (13) | 1 | (5) | 4 | (22) | 0.17 |
| 2 –n (%) | 33 | (87) | 19 | (95) | 14 | (78) | 0.17 |

eGFR, estimated glomerular filtration rate; IQR, interquartile range; LDH, lactate dehydrogenase; RBC, red blood cells.

[a]Connective tissue disease (systemic lupus erythematosus in 3, mixed connective tissue disease in 1 and anti-synthetase syndrome in 1), immunosuppressive treatment, and hepatitis C.

[b]Pregnancy and surgery.

[c]Including transient ischemic attack, stroke, seizure and altered mental status.

[d]Khelaff hemorrhagic score.

[e]Defined by hemoglobin <12 g/dL for women and <13 g/dL for men.

[f]Missing data (total/immediate diagnosis/delayed diagnosis): reticulocytes (12/1/11), schistocytes (11/1/10), LDH (14/3/11), bilirubin (11/4/7).

[g]French score is a prediction score for TTP diagnosis, after ruling out intravascular disseminated coagulation, cancer, and transplantation situation. French score = 2 is highly predictive of TTP diagnosis.

Most neurological events occurred within 1 week after the first hospital evaluation. Of note, among 16 patients (42%) who suffered from at least one neurological event, the first occurred before TPE initiation in 12 and during TPE in 4. Details regarding neurological symptoms are

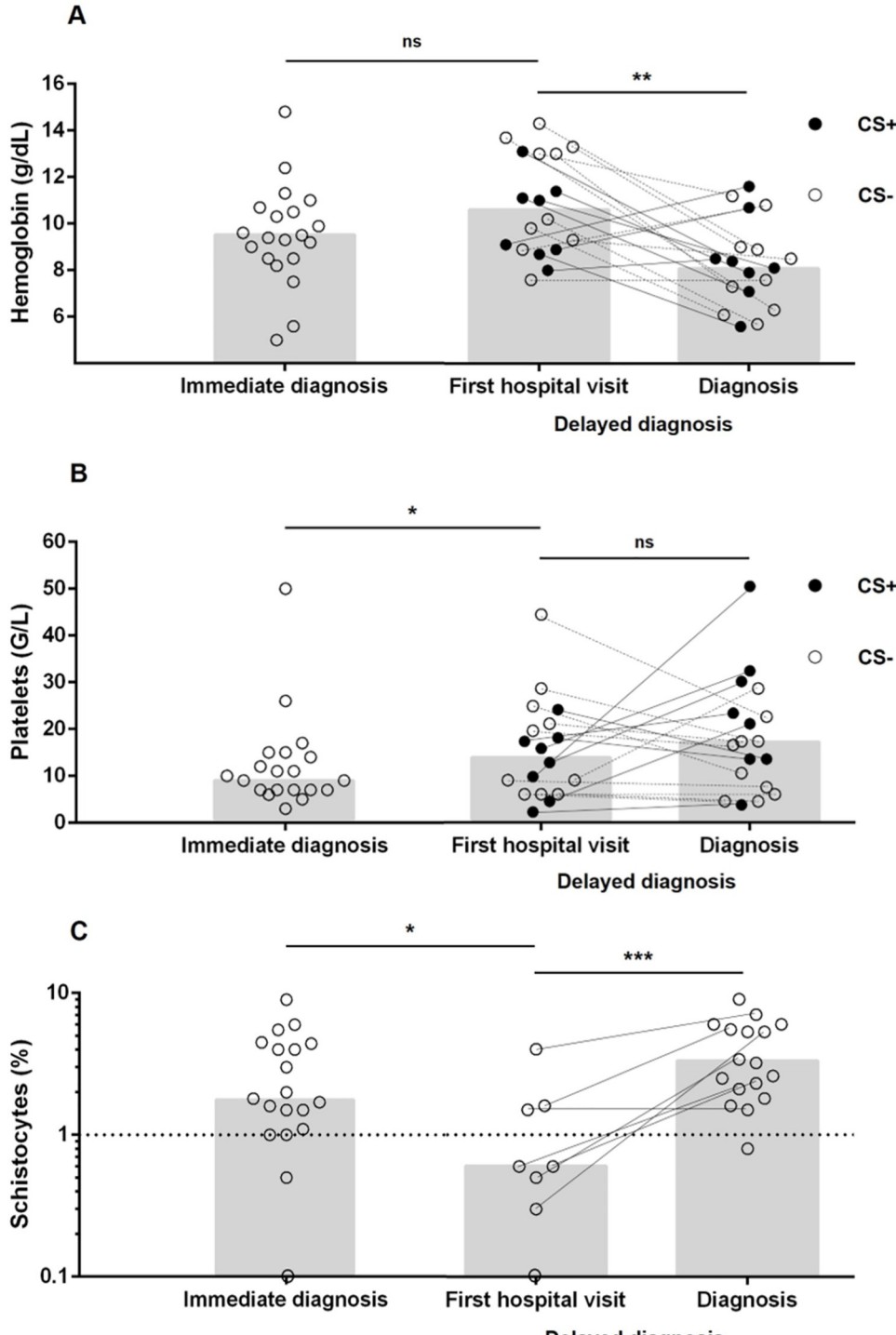

**Fig 1. Comparison and evolution of biological features.** Hemoglobin level (A), platelet count (B), and schistocytes count (C) before therapeutic plasma exchange in patients with immediate (left) or delayed diagnosis (right). Black dots depict patients that received corticosteroids (CS) before iTTP diagnosis.

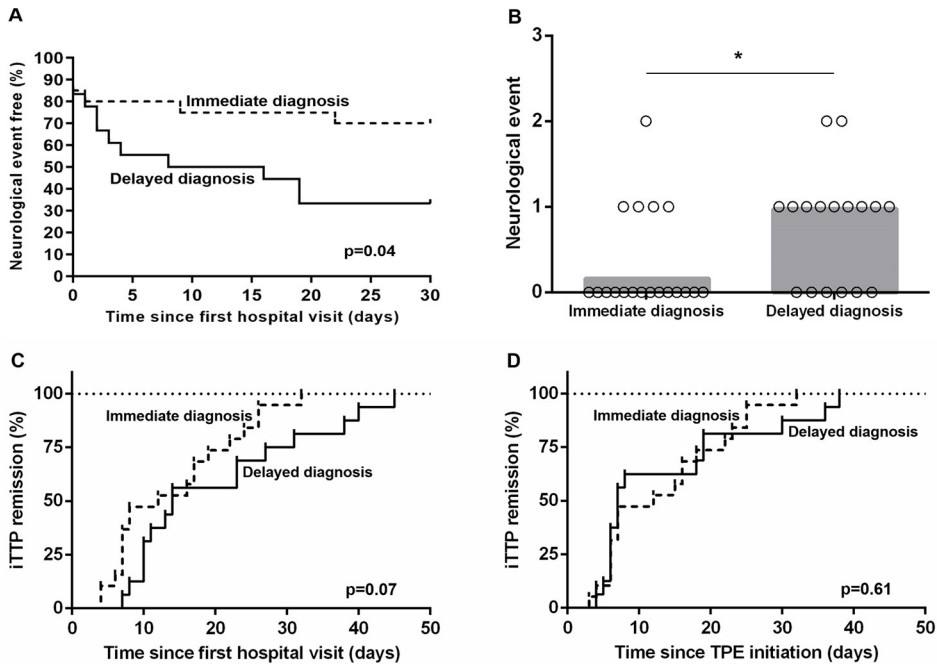

**Fig 2. Comparison of neurological outcome and response to therapy.** Neurological event free survival (A), median number of neurological events per patient (B), and cumulative incidence of remission (C: from first hospital visit; D: from plasma exchange initiation).

available in Table 2. At hospital discharge, 2 patients had neurological sequelae, both in the delayed diagnosis group: 1 had a brachio-facial motor palsy with aphasia and 1 had pyramidal tetraparesis with tremor, both of whom were still present at last follow-up. The mortality rate was 5%, with 1 death in each group: 1 patient with a delayed diagnosis died due to refractory iTTP 20 days after TPE initiation, and 1 patient with immediate diagnosis died <24h after admission, the same day of TPE initiation.

**Table 2. Comparison of neurological outcomes from disease onset to hospital discharge.**

| | Total | | Immediate diagnosis | | Delayed diagnosis | | *P-value* |
|---|---|---|---|---|---|---|---|
| | (n = 38) | | (n = 20) | | (n = 18) | | |
| Neurological manifestations[a] | | | | | | | |
| Headache–n (%) | 16 | (42) | 7 | (35) | 9 | (50) | 0.51 |
| Neurological event–n (%) | 18 | (47) | 6 | (30) | 12 | (67) | **0,049** |
| Stroke/TIA [n][a] | 15 | [14] | 3 | [3] | 12 | [11] | **0,006**[b] |
| Seizure [n][a] | 6 | [4] | 3 | [1] | 3 | [3] | 0.33[b] |
| Altered mental status [n][a] | 6 | [6] | 4 | [4] | 2 | [2] | 0.66[b] |
| Unfavorable outcomes | | | | | | | |
| Neurological sequelae–n (%) | 2 | (5) | 0 | (0) | 2 | (11) | 0.22 |
| Death–n (%) | 2 | (5) | 1 | (5) | 1 | (6) | 1.0 |
| Total–n (%) | 4 | (10) | 1 | (5) | 3 | (17) | 0.33 |

TIA, transient ischemic attack.

[a]Number of neurological events [number of patients].

[b]Fischer's exact test performed using the number of patients.

**Table 3. Comparison of secondary outcomes.**

| | Total | | Immediate diagnosis | | Delayed diagnosis | | P-value |
|---|---|---|---|---|---|---|---|
| | (n = 38) | | (n = 20) | | (n = 18) | | |
| Length of stay[a] | | | | | | | |
| ICU–days (IQR) | 7 | (3–13) | 7 | (3–14) | 7 | (5–11) | 0.97 |
| Hospital–days (IQR) | 28 | (22–35) | 24 | (17–34) | 30 | (27–39) | **0.02** |
| iTTP treatment | | | | | | | |
| Fresh frozen plasma–n (%) | 5 | (13) | 3 | (15) | 2 | (11) | 1.0 |
| Plasma exchange–n (%) | 38 | (100) | 20 | (100) | 18 | (100) | 1.0 |
| TPE cycle per patient–n (IQR)[b] | 19 | (10–29) | 16 | (11–26) | 25 | (10–33) | 0.3 |
| Corticosteroids–n (%) | 32 | (84) | 16 | (80) | 16 | (89) | 0.76 |
| Rituximab–n (%) | 29 | (76) | 14 | (70) | 15 | (83) | 0.56 |
| Caplacizumab–n (%) | 4 | (11) | 3 | (15) | 1 | (6) | 0.61 |
| Erythrocyte transfusion–n (%) | 23 | (61) | 9 | (45) | 14 | (78) | 0.05 |
| ET per patient–n (IQR) | 1.5 | (0–2) | 0 | (0–2) | 2 | (1–2) | 0.07 |
| Other treatment | | | | | | | |
| Intravenous Ig–n (%) | 6 | (16) | 0 | (0) | 6 | (33) | **0.007** |
| TPO analogue–n (%) | 1 | (3) | 0 | (0) | 1 | (6) | 0.47 |
| Platelet transfusion–n (%) | 13 | (34) | 4 | (20) | 9 | (50) | 0.09 |
| PT per patient–n (IQR) | 0 | (0–1) | 0 | (0–0) | 0.5 | (0–1) | 0.06 |

ICU, intensive care unit; iTTP, immune-mediated thrombotic thrombocytopenic purpura; TPE, therapeutic plasma exchange; ET, erythrocyte transfusion; Ig, Immunoglobulin; TPO, thrombopoietin; PT, platelet transfusion; IQR, interquartile range.

[a]Time since first hospital visit.

[b]the two patients who died were excluded.

We display a comparison of patients' secondary outcomes in Table 3. The hospital length of stay was longer in patients with a delayed diagnosis (24 vs 30 days, p = 0.02). The median time from TPE initiation to remission was 7 days (IQR 7–19) in the overall cohort, without difference between the 2 groups. Expectedly, the median time from first hospital visit to remission tended to be longer in patients with delayed diagnosis (12 vs 14 days, p = 0.07) (Fig 2C and 2D). The use of TPE, corticosteroids, rituximab, and caplacizumab was similar in the 2 groups.

## Clinical-biological determinants of diagnostic delay

Comparison of baseline characteristics of patients with or without delayed diagnosis is reported in Table 1. Patients with delayed diagnosis more frequently had at least one underlying condition that could cause thrombocytopenia [6 (33%) vs 1 (5%), p = 0.04]: 1 patient (3%) had hepatitis C, 5 patients (14%) had a connective tissue disease, and 2 patients (3%) received an immunosuppressive treatment. One patient had 2 underlying conditions at the same time (connective tissue disease and immunosuppressive treatment). We identified a potential precipitating factor in 3 patients (8%), all in the delayed diagnosis group: 2 had recent surgery (5%) and 1 was pregnant (3%). As for clinical manifestations, we observed no significant difference. As for biological findings, patients with delayed diagnosis more frequently presented with a normal hemoglobin level (33% vs 5%, p = 0.04). Haptoglobin level and schistocyte count were less frequently ordered in the delayed diagnosis group (100% vs 44%, p<0.01 and 95% vs 44%, p<0.01, respectively). By contrast, a higher proportion of patients of the delayed diagnosis group had undergone bone marrow examination (14/18 [78%] vs 5/20 [25%],

p<0.01). Among 27 patients with a schistocytes count performed at their first hospital visit, 2 patients (7%) had no detectable schistocytes (i.e., 0%), 5 patients (19%) had rare schistocytes (i.e., 0.1–1%) and 20 patients (74%) had schistocytes count equal or above 1%. Moreover, 5/7 patients (71%) with a schistocytes count below 1% at first hospital visit versus 3/20 (15%) patients above 1% had a diagnosis delay (p = 0.01). Furthermore, schistocytes count at first hospital visit correlated negatively with time to diagnosis (r = -0.43 [95% CI: -0.70 to -0.04], p = 0.03).

## Discussion

Like others acute TMA, iTTP is a rare deadly disease that can pose difficult diagnostic and therapeutic dilemmas. The life-saving efficacy of plasma therapy is known for decades but our understanding of iTTP has only been revolutionized in early 2000, when severe acquired ADAMTS13 deficiency was recognized as the key pathogenic event and led to dissecting the TTP/HUS spectrum [14–16]. However, given the variety of TMA causes and the delay required to obtain ADAMTS13 activity measurement, the decision to recourse to iTTP treatment (TPE, corticosteroids, caplacizumab, rituximab. . .) in a patient with TMA still relies on simple clinical-biological assessment [17–19]. Contemporary data regarding diagnostic delay in iTTP are rare [11, 12]. Our study aimed to assess the frequency, clinical consequences and determinants of diagnostic delay in iTTP.

According to the French score, the probability of TTP diagnosis was >70% in all of our patients at the first hospital evaluation. Yet, almost half of them experienced diagnostic delay. TTP probability was even > 94% in 78% of patients with delayed diagnosis. Recent studies also showed that early diagnosis and treatment remain challenging in iTTP. In the French TMA registry, 20% of iTTP patients were initially misdiagnosed [12]. In a recent Canadian study, TPE initiation delay was >24h in 24% of patients with suspected new onset or relapsing iTTP [11]. Our second observation is that neurological events occurred much more frequently in patients with delayed diagnosis (67% vs 30%). Until recently, little data were available regarding the clinical impact of therapeutic delay in iTTP and mostly focused on the risk of death. In 2017, Grall et al. reported that, unexpectedly, misdiagnosed iTTP patients (84 out of 423) had similar survival rate (12–13%) [12]. In 2020, Sawler et al. analyzed 80 iTTP episodes in 61 patients and found that delayed TPE (>24h) tended to be associated with a higher risk of death that did not reach statistical significance (aHR = 1.40, 95% CI = 0.20–9.79) [11]. They also reported that delayed TPE (>24h) were only associated with a non-significant increase in thrombotic risk (aHR 2.9, 95% CI 0.6–12.8). Our data are consistent with historical knowledge regarding iTTP course and demonstrate the deleterious neurological impact of a non-immediate diagnostic. Overall, despite reassuring data regarding mortality, expedite diagnosis and treatment must remain a key objective in iTTP [17–19]. Recent data show that a significant proportion of iTTP patients report persistent cognitive symptoms and depression [20–22]. Furthermore, a correlation was found between initial neurological manifestations, cerebral MRI scan abnormalities and long-term cognitive performances [23]. Further studies are needed to determine how initial diagnostic and therapeutic management impact the long-term neuropsychiatric burden of iTTP patients.

In our patients, the most frequent misdiagnosis was immune thrombocytopenia and Evans syndrome, consistent with Grall et al. observation [12]. Thus, despite the focus of most literature on the distinction between iTTP and other TMA [2, 13, 24], this issue does not seem to contribute to diagnosis delay in real life. Instead, we suggest that the 2 key factors contributing to diagnostic delay are the failure to either suspect or demonstrate TMA. In almost 80% of cases, first hospital visit occurred in areas where TMA is an exceptional diagnosis, mostly in

the emergency room (ER), suggesting that any educational program should include hospitalists, but, most importantly, emergency physicians. From a clinical standpoint, patients with or without diagnostic delay had similar disease manifestations at first hospital evaluation. At that time, overt neurological and/or cardiac damage was rare. However, several symptoms tended to be neglected despite being hardly attributable to biological abnormalities alone (headache, nausea, abdominal pain, intense fatigue) and probably reflected early sign of organ dysfunction. Expectedly, the decisive biological finding was severe thrombocytopenia. Anemia, rarely important at presentation, was less prevalent in patients with diagnostic delay, consistently with Grall et al. findings [12]. Furthermore, a potential alternative cause of thrombocytopenia (connective tissue disease, hepatitis C infection, immunosuppressive therapy and heparin exposure) or a precipitating factor (pregnancy, surgery) was more frequently found in patients with diagnostic delay. Although a personal history of auto-immunity, pregnancy, and surgery should have led to suspect iTTP, among others immune-mediated hypothesis (catastrophic anti-phospholipid syndrome, immune thrombocytopenia. . .), these parameters likely confused clinical reasoning [12, 25, 26]. Regarding biological assessment, initial hemolysis work-up and schistocytes count were missing in more than 60% of cases with delayed diagnosis. By contrast, these patients were more frequently subjected to bone marrow examination. These observations clearly reflect the failure of clinicians to suspect TMA. However, demonstrating TMA may also be challenging. Initial schistocytes counts were below the 1% threshold in 26% of cases, which was significantly associated with a delayed diagnosis. The poor sensitivity of this criterion for the early diagnosis of iTTP is also apparent in Grall et al study, in which 38% of patients were below this threshold initially [12]. Conversely, we show that when measured, haptoglobin was consistently <0.1g/L at the first hospital visit. Thus, a normal haptoglobin level makes the iTTP diagnosis highly unlikely. Yet, thrombocytopenia associated with undetectable haptoglobin can be seen in other setting than TMA (i.e. Cobalamin deficiency, Evans syndrome, hematoma resorption, intramedullary hemolysis, paroxystic nocturnal hematuria, malaria, hepatic failure. . .) [27]. The identification of schistocytes on a blood smear, the cornerstone of TMA diagnosis, critically depends on the biologist's experience [28]. Schistocytes can be found in many conditions such as neoplasia, severe infection, hematological stem-cell transplant, cobalamin deficiency, and certain drugs intake [29]. Schistocyte count can be considered normal if <0.2% in healthy people, 0.6% in patients with renal failure, and 0.48% in patients with normally functioning prosthetic valves [30]. The official cut-off for the schistocytes count retained by the International Council for Standardization in Hematology in TMA is 1% (except in TMA associated with hematopoietic stem-cell transplant whose cut-off is 4%) [30]. The fact that schistocytes represent the main morphological red blood cell abnormality on the blood film increases its clinical significance [29]. Our data show that, in the appropriate clinical setting, even a low level of isolated schistocytosis (0.1–1%) can play a crucial role in the early diagnosis of iTTP. They illustrate the value of repeated schistocytes count but also suggest that clinicians should be able to order TPE in patients without significant schistocytes. Moreover, troponin level, an excellent marker of organ ischemia in iTTP [31], should be more systematically measured in patients with unexplained profound thrombocytopenia and hemolysis, even with no detectable schistocytes. Of note, blood smear analysis is systematically performed in patients with thrombocytopenia, to rule out platelet aggregates. How the implementation of health electronic records' red flags, or systematic schistocytes count and haptoglobin measurement, could improve the early diagnosis of iTTP among patients presenting to the ER with severe thrombocytopenia deserves to be investigated.

Our study has some strengths and limitations. One of its strengths is that, unlike tertiary care or multicenter registries, we were able to collect extensive data regarding the early clinical-biological course of our patients, including at their very first local hospital visit, which

allowed us to capture the deleterious neurological consequences of diagnostic delay. One of the limitations of our study is the limited number of patients, which is expected given the rarity of the disease. Our work also suffers the usual limitations of retrospective studies. We cannot exclude that some undiagnosed iTTP with a fatal outcome were missed, which would only reinforce our findings regarding the deleterious impact of misdiagnosis. Another limitation is missing biological data. However, this issue reflects real life practices and allowed us to identify avenues for clinical practice improvement.

## Conclusion

Even in the ADAMTS13 era, the early identification and treatment of iTTP remain challenging with clinically relevant impact. Our study suggests that beyond the distinction between iTTP and other TMA, the main issues for clinical practice improvement is to increase the awareness of first line physicians regarding the early clinical-biological presentation of iTTP.

## Supporting information

**S1 File. Dataset.**
(XLSX)

## Acknowledgments

The authors would like to thank Dr Julie Chevalier and Dr Fanny Giroux from the Hospital Center Bretagne Atlantique in Vannes, France, for the help brought to data collection.

## Author Contributions

**Conceptualization:** Antoine Néel.

**Data curation:** Arthur Renaud, Aurélie Caristan, Amélie Seguin, Christian Agard, Gauthier Blonz, Emmanuel Canet, Marion Eveillard, Pascal Godmer, Julie Graveleau, Marie Lecouffe-Desprets, Hervé Maisonneuve, François Perrin, Mohamed Hamidou, Antoine Néel.

**Formal analysis:** Arthur Renaud, Antoine Néel.

**Methodology:** Arthur Renaud, Aurélie Caristan, Mohamed Hamidou, Antoine Néel.

**Software:** Arthur Renaud.

**Supervision:** Mohamed Hamidou, Antoine Néel.

**Validation:** Arthur Renaud, Amélie Seguin, Christian Agard, Gauthier Blonz, Emmanuel Canet, Marion Eveillard, Pascal Godmer, Julie Graveleau, Marie Lecouffe-Desprets, Hervé Maisonneuve, François Perrin, Mohamed Hamidou, Antoine Néel.

**Writing – original draft:** Arthur Renaud.

**Writing – review & editing:** Arthur Renaud, Aurélie Caristan, Amélie Seguin, Christian Agard, Gauthier Blonz, Emmanuel Canet, Marion Eveillard, Pascal Godmer, Julie Graveleau, Marie Lecouffe-Desprets, Hervé Maisonneuve, François Perrin, Mohamed Hamidou, Antoine Néel.

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
