## [Decision Letter · Decision Letter 0]

30 Jun 2021

PONE-D-21-15104

Deleterious neurological impact of diagnostic delay in immune-mediated thrombotic thrombocytopenic purpura

PLOS ONE

Dear Dr. Renaud,

Thank you for submitting your manuscript to PLOS ONE. After careful consideration, we feel that it has merit but does not fully meet PLOS ONE’s publication criteria as it currently stands. Therefore, we invite you to submit a revised version of the manuscript that addresses the points raised during the review process.

We look forward to receiving your revised manuscript.

Kind regards,

Tai-Heng Chen, M.D.

Academic Editor

PLOS ONE

Journal Requirements:

2. Thank you for your submission, and your ethics statement reading,

"This study is in accordance with the Declaration of Helsinki, and the French Data

Protection Authority and Legislation (MR003 reference methodology). No change in the

current clinical practice and no randomization were performed. As it was a

retrospective study, according to the French legislation (articles L.1121-1 paragraph 1

and R1121-2, Public health code), approval of the ethics committee was not needed to

use data for this study."

At this time, we request that you please indicate whether all data was anonymized and/or aggregated prior to your access and analysis, or whether authors had access to identifying patient information. Thank you for your attention to this request.

Reviewers' comments:

Reviewer's Responses to Questions

**Comments to the Author**

1. Is the manuscript technically sound, and do the data support the conclusions?

Reviewer #1: Yes

Reviewer #2: Partly

2. Has the statistical analysis been performed appropriately and rigorously? 

Reviewer #1: Yes

Reviewer #2: Yes

3. Have the authors made all data underlying the findings in their manuscript fully available?

Reviewer #1: Yes

Reviewer #2: Yes

4. Is the manuscript presented in an intelligible fashion and written in standard English?

Reviewer #1: Yes

Reviewer #2: No

5. Review Comments to the Author

Reviewer #1: In this manuscript, the authors investigated the effects of diagnostic delay on neurological in patients with iTTP. Based on their results, Renaud et al. concluded that diagnostic delay had a significant impact on neurological outcome. Although the results of this study add a little new information to current literatures, the retrospective nature and small number of cases make it difficult to draw a definite conclusion that “diagnostic delay has an impact on neurological outcomes in iTTP”. My comments:

1. Why the cutoff value of 24h was used to group classification ? Please add the number of patients in group 1 and group 2.

2. Please add references for the definition of iTTP.

3. Please add details on the treatment of iTTP.

4. How to define “brain dysfunction” and “brain dysfunction free survival” ? and related references should be provided.

5. The authors did not observe the effect of diagnostic delay on mortality. Why ?

Reviewer #2: The current article from Renaud and colleagues investigates the effect and reasonings of diagnostic delay of TTP and neurological outcomes in affected patients. Outlining the diagnostic during the differential diagnosis between TTP and other diseases is of great interest; however, several significant questions are raised. Moreover, the manuscript at this stage provides only confirmatory results that >24h delay markedly increase negative consequences. The delay in diagnosis directly correlates with the delay in a specific treatment, and the consequences of the latter are reported by Sawler et al., 2020

Major comments:

1. Although the diagnosis establishment is essential, how does it correlate with the plasma exchange therapy start? Are there patients from the group with the immediate diagnosis but delayed plasma transfusions?

2. How was the cut-off for the delayed and immediate diagnosis determined?

3. In what number of patients were the ADAMTS13 levels determined before or shortly after the plasma exchange therapy?

4. The authors state that different therapeutic approaches were used. Please, specify what treatment patients received in different groups and their counts.

5. Several prediction scores were developed to assess TTP risks. Did clinicians use them to evaluate the probability of TTP? It is interesting to check if they are significantly different between derived groups.

6. The Clinical-biological determinants of diagnostic delay section raise several comments:

a. Although the authors state that the group has statistically different PLT counts, one would argue that both levels might be considered as severe thrombocytopenia.

b. Mixing values from total counts and delayed group when analyzing schistocytes strongly confuses the reader

Minor comments:

1. Please, specify the ADAMTS13 assay type used in the study.

2. Table 1 and Table 2 content is not immediately straightforward for the reader. Two subcolumns are reported in each section; however, it’s not apparent what is reported in each one and in what units. For example, Women 26 (68). I strongly recommend reformating the tables to include all the numbers

3. The manuscript requires proof-reading by an English native person

6. PLOS authors have the option to publish the peer review history of their article (what does this mean?). If published, this will include your full peer review and any attached files.

Reviewer #1: No

Reviewer #2: No

---

## [Author Response · Author response to Decision Letter 0]

25 Aug 2021

Dear reviewers and academic editor,

I have uploaded a letter in a separate file entitled "Response to Reviewers" that addresses each specific question and comment raised in the decision letter.

Best regards,

Arthur RENAUD

---

## [Decision Letter · Decision Letter 1]

29 Sep 2021

PONE-D-21-15104R1Deleterious neurological impact of diagnostic delay in immune-mediated thrombotic thrombocytopenic purpuraPLOS ONE

Dear Dr. Renaud,

Thank you for submitting your manuscript to PLOS ONE. After careful consideration, we feel that it has merit but does not fully meet PLOS ONE’s publication criteria as it currently stands. Therefore, we invite you to submit a revised version of the manuscript that addresses the points raised during the review process.

We look forward to receiving your revised manuscript.

Kind regards,

Tai-Heng Chen, M.D.

Academic Editor

PLOS ONE

Journal Requirements:

Reviewers' comments:

Reviewer's Responses to Questions

**Comments to the Author**

1. If the authors have adequately addressed your comments raised in a previous round of review and you feel that this manuscript is now acceptable for publication, you may indicate that here to bypass the “Comments to the Author” section, enter your conflict of interest statement in the “Confidential to Editor” section, and submit your "Accept" recommendation.

Reviewer #3: (No Response)

2. Is the manuscript technically sound, and do the data support the conclusions?

Reviewer #3: Partly

3. Has the statistical analysis been performed appropriately and rigorously? 

Reviewer #3: Yes

4. Have the authors made all data underlying the findings in their manuscript fully available?

Reviewer #3: Yes

5. Is the manuscript presented in an intelligible fashion and written in standard English?

Reviewer #3: Yes

6. Review Comments to the Author

Reviewer #3: ABSTRACT

> Unexpectedly, recent studies suggest that a slight delay in TPE initiation has no significant impact on patients’ outcome.

Some other expressions like "However, the exact impact of a slight delay in TPE initiation on the subsequent patients’ outcome is still controversial” would be better.

> delayed (>24h, group 1) and immediate diagnosis (≤24h, group 2).

24 hours from when? From the clinical onset, or from the first hospital visit?

> Conclusion: Diagnostic delay is highly prevalent in iTTP, with a significant impact on neurological outcome.

The shown results imply that the occurrence of stroke/TIA during hospitalization may be slightly higher in “delayed” group, whereas the occurrence of irreversible neurological sequelae was not significantly different (p=0.22) between the groups. Thus, this conclusion is misleading. The swift performance of TPE may suppress the occurrence of stroke/TIA, but it may not improve the neurological outcome.

> Interdisciplinary efforts are necessary in order to increase first-line physicians’ awareness regarding the early clinical-biological presentation of iTTP.

What is the author’s specific message regarding the suggested factors of delayed diagnosis? Please describe more specifically. Is there a supporting data to say that the “unawareness” among the first-line physicians of the disease concept of “TTP” caused the delayed diagnoses in your cohort?

METHODS

Graph Pad Prism software 6.0 >> GraphPad Prism software 6.0

Please add the information about the manufacturer of GraphPad Software.

e.g., (GraphPad Software, Inc., San Diego, CA)

I cannot find the criteria and rationale to divide the cohort to the “delayed” group (>24h) and “immediate” group (<24h) in the METHODS section. From when the “24 hours” was counted? From the clinical onset? Why the cutoff time of 24 hours was set? Did previous studies support the cutoff at 24 hours of TPE initiation from onset to influence the clinical outcome?

RESULTS

> Clinical- biological determinants of diagnostic delay

How about performing a multiple regression analysis by using the required days from onset to TPE initiation as the dependent variable? Currently, the authors only performed univariate analyses to search for the determinants of diagnostic delay.

TABLES

Table 1:

How were the data regarding the prevalence of fever at the first hospital visit in the two groups?

> no. (%)

What is “no”? Maybe, “n” would be better, if the authors intended to show the numbers.

> points (IQR)

Are the shown values median or mean?

(N=38) (N=20) (N=18) >> (n=38) (n=20) (n=18)

“N” and “n” have different meanings.

Table 2:

> Table 2. Comparison of neurological outcomes from disease onset to hospital discharge

I think the table should compare the neurological outcomes from “TPE initiation” to hospital discharge, because this section intends to show the “Deleterious neurological impact of diagnostic delay”. If they include the time “from disease onset” in this table, the numbers overlap with those in Table 1.

> a Number of neurological events [number of patients].

Which of the “number of events” or “number of patients” was used to calculate the p-values by performing chi square or Fisher test?

> Stroke/TIA [n] a

Please separately show the numbers for “stroke” and “TIA”. Did skilled neurologists diagnosed the TIA conditions? Were all episodes of stroke confirmed by brain MRI scan?

Table 3:

> TPE per patient – no. (IQR)b

Maybe, TPE “cycles” per patient?

7. PLOS authors have the option to publish the peer review history of their article (what does this mean?). If published, this will include your full peer review and any attached files.

Reviewer #3: No

---

## [Author Response · Author response to Decision Letter 1]

19 Oct 2021

Dear Academic Editor and Reviewer,

I addressed each point raised in the Decision Letter in a separate file untitled "Response to Reviewer". All modifications mentioned in this letter have been highlighted in another separate file untitled "Revised Manuscript with Track Changes".

I hope this responds to your questions and comments.

Best regards,

Arthur RENAUD

---

## [Decision Letter · Decision Letter 2]

5 Nov 2021

Deleterious neurological impact of diagnostic delay in immune-mediated thrombotic thrombocytopenic purpura

PONE-D-21-15104R2

Dear Dr. Renaud,

We’re pleased to inform you that your manuscript has been judged scientifically suitable for publication and will be formally accepted for publication once it meets all outstanding technical requirements.

Kind regards,

Massimo Cugno, M.D.

Academic Editor

PLOS ONE

Additional Editor Comments (optional):

Reviewers' comments:

Reviewer's Responses to Questions

**Comments to the Author**

1. If the authors have adequately addressed your comments raised in a previous round of review and you feel that this manuscript is now acceptable for publication, you may indicate that here to bypass the “Comments to the Author” section, enter your conflict of interest statement in the “Confidential to Editor” section, and submit your "Accept" recommendation.

Reviewer #1: All comments have been addressed

Reviewer #3: All comments have been addressed

2. Is the manuscript technically sound, and do the data support the conclusions?

Reviewer #1: Yes

Reviewer #3: Yes

3. Has the statistical analysis been performed appropriately and rigorously? 

Reviewer #1: Yes

Reviewer #3: Yes

4. Have the authors made all data underlying the findings in their manuscript fully available?

Reviewer #1: Yes

Reviewer #3: Yes

5. Is the manuscript presented in an intelligible fashion and written in standard English?

Reviewer #1: Yes

Reviewer #3: Yes

6. Review Comments to the Author

Reviewer #1: All of my comments have been addressed by the authors. The manuscript could be accepted for publication.

Reviewer #3: Thank you for the thorough modifications.

I think the authors have addressed all my concerns correctly.

7. PLOS authors have the option to publish the peer review history of their article (what does this mean?). If published, this will include your full peer review and any attached files.

Reviewer #1: No

Reviewer #3: No

---

## [Editor Report · Acceptance letter]

10 Nov 2021

PONE-D-21-15104R2 

Deleterious neurological impact of diagnostic delay in immune-mediated thrombotic thrombocytopenic purpura 

Dear Dr. Renaud:

I'm pleased to inform you that your manuscript has been deemed suitable for publication in PLOS ONE. Congratulations! Your manuscript is now with our production department. 

Kind regards, 

on behalf of

Professor Massimo Cugno 

Academic Editor

PLOS ONE